



**1 Comprehensive evaluation of typical planetary boundary**
**2 layer (PBL) parameterization schemes in China. Part II:**
**3 Influence of uncertainty factors**

Wenxing Jia[1], Xiaoye Zhang[1,2*], Hong Wang[1], Yaqiang Wang[1], Deying Wang[1], Junting Zhong[1],
Wenjie Zhang[1], Lei Zhang[1], Lifeng Guo[1], Yadong Lei[1], Jizhi Wang[1], Yuanqin Yang[1], Yi Lin[3]
[1]State Key Laboratory of Severe Weather & Key Laboratory of Atmospheric Chemistry of CMA,
Chinese Academy of Meteorological Sciences, Beijing, 100081, China
[2]Center for Excellence in Regional Atmospheric Environment, IUE, Chinese Academy of Sciences,
Xiamen, 361021, China
[3]Key Laboratory for Mesoscale Severe Weather, Ministry of Education, and School of Atmospheric
Sciences, Nanjing University, Nanjing, China
Correspondence to: X. Zhang (xiaoye@cma.gov.cn)





**Abstract.** This study focuses on the uncertainties that influence numerical simulation results of meteorological fields (including horizontal resolution, vertical resolution, near-surface scheme, initial and boundary conditions, underlying surface update, and update of model version). By further evaluating and analyzing the uncertainty factors, it is expected to provide relevance for those scholars who are devoted to factor analysis, in order to make the results closer to the observed values. In this study, a total of 12 experiments are set up to analyze the effects of the uncertainties mentioned above, and the following conclusions are drawn: (1) Horizontal resolution has a greater effect than vertical resolution. (2) The simulated effects of temperature and wind speed in the near-surface scheme are smaller than those in the PBL scheme. (3) The initial and boundary conditions of different products have the most remarkable effect on relative humidity, and the simulation results of EC data are the best. (4) The updates with urban and water bodies as the underlying surface have a more significant contribution to the meteorological fields, especially on temperature. (5) The update of the model version does not necessarily optimize the model results. In general, the configuration of uncertainties needs to be considered comprehensively according to what you need in order to obtain the best simulation results.

## Introduction

The key factor for the accurate simulation of near-surface meteorological elements and planetary boundary layer (PBL) structure is the PBL parameterization scheme. Part I has discussed the impact of the PBL scheme in detail from the mechanism, and assessed the applicability of the PBL scheme for different parameters in different regions and seasons. However, there are still many uncertainties in the model that can affect the forecasts and results. The model settings used by different scholars exhibit differences in the simulation results. For example, the horizontal and vertical resolutions are essential for model settings. Horizontal resolution, as a critical factor, must be considered in all models, whether they are macroscale earth system models (ESMs), climate models, mesoscale regional models, or microscale fluid models. Constrained by computational resources, a horizontal resolution of about 100~250 km is used for ESMs models (e.g., Coupled Model Intercomparison Project phase 6, CMIP6 model) (D. Li et al., 2022; Taylor, Stouffer, & Meehl, 2012). The horizontal resolution of climate models typically ranges from 50 to 25 km (e.g., Flexible Global Ocean-Atmosphere-Land System Finite-Volume version 3, FGOALS-f3 model)(J. Li et al., 2021). The



horizontal resolution of mesoscale weather models (e.g., The Global/Regional Assimilation
Prediction System, GRAPES model, Weather Research and Forecasting, WRF model)(García-
García, Cuesta-Valero, Beltrami, González-Rouco, & García-Bustamante, 2022; Ma et al., 2018)
can be as fine as 1 km. The microscale fluid models can have a horizontal resolution of less than
100 m (e.g., Large eddy simulation, LES model)(Zhou, Zhu, & Xue, 2017). Studies have shown
that the interaction between large- and small-scales is influenced by resolution, with finer resolution
allowing for better characterization of underlying surface features and extreme
events(Rummukainen, 2016; Singh et al., 2021), and also impacting future climate
predictions(Chang et al., 2020; Roberts et al., 2020; Small et al., 2014). The use of PBL scheme is
usually in coarse resolution models, which can lead to additional errors since these schemes are
developed for flat terrain conditions(Weigel, Chow, & Rotach, 2007).
Finer vertical resolution can better capture changes in PBL structure, which can also have an impact
on mass transport(Menut, Bessagnet, Colette, & Khvorostiyanov, 2013; O'Dea et al., 2017; Teixeira,
Carvalho, Tuccella, Curci, & Rocha, 2016), especially on the accuracy of wind resources (Tolentino,
Rejuso, Inocencio, Ang, & Bagtasa, 2016). In addition, horizontal and vertical resolutions can cause
spurious gravity waves and increase model errors(Nolan & Onderlinde, 2022). Although higher
resolution is better, there is no doubt that it is computationally expensive. Whether the use of finer
resolution will bring significant improvement to the model results deserves further discussion.
Different PBL schemes are combined with the different near-surface (N-S) schemes, both of which
are crucial to the simulation results of the meteorological fields. For instance, the MYJ scheme (i.e.,
PBL scheme) can only couple the Eta scheme (i.e., N-S scheme), while the BL scheme (i.e., PBL
scheme) can couple both the MM5 and the Eta schemes (i.e., N-S scheme). The N-S scheme is
pivotal for mesoscale numerical simulation, especially for fine numerical forecasting(Y. Li, Gao,
Lenschow, & Chen, 2010). Then, to figure out which scheme has a greater impact on the
meteorological field will help to make targeted improvements to the forecasts in the future. In
addition, the lag of the underlying surface data can also affect the simulation results of the
meteorological fields, especially for large cities with relatively rapid urbanization (Qian et al., 2022).
The European Center for Medium-Range Weather Forecasting (ECMWF, hereafter referred to as
EC) has concluded that the steady progress in numerical forecasting over the last 30 years is mainly
attributed to improvements in the forecast models themselves, the application of more observations
and the development of data assimilation techniques(Magnusson & Källén, 2013). Among them,





the performance of the forecast model depends largely on the model resolution, the accuracy of the
finite difference method, and the representativeness of the physical process parameterization
scheme. Different initial fields also influence the model results due to different observational data,
quality control methods, assimilation schemes, and bias correction methods adopted for different
reanalysis data(Ma et al., 2021).
Finally, we also have to take the update of the model version into account. With model versions
being updated, many parameterization schemes are more or less updated(Morichetti et al., 2022).
However, under the circumstance that the updates are not disclosed in scientific and technical reports
or papers, we need to dig into them from the code itself. In reality, simulation results will be likely
to vary from scholar to scholar because of different model versions they choose. Consequently, it is
necessary to adopt a control variable approach when discussing the impact of model version updates.
Instead of updating all parameterization schemes, only by updating the ones we are concerned with
can the uncertainty arising from version updates can be quantified.
These aforementioned uncertainties have been studied by scholars individually, but few scholars
have been able to synthesize and analyze these factors. In this part, each of these uncertainties will
be analyzed and discussed, and the factors with more significant effects will be selected for reference
in that identifying which factors besides the PBL scheme are critical to the simulation of
meteorological fields makes all the difference.

## 2. Data and Methodology

### 2.1 Data

#### 2.1.1 Reanalysis Data

*Final (FNL) reanalysis data.* The National Centers for Environmental Prediction (NCEP)
global final (FNL) reanalysis data are based on the 6 h temporal resolution (i.e., 00:00 (08:00),
06:00 (14:00), 12:00 (18:00), 18:00 (02:00) UTC (BJT)) by the Global Data Assimilation
System (GDAS) with a resolution of $1° \times 1°$ or $0.25° \times 0.25°$. This product continuously
collects observational data from the Global Telecommunications System (GTS) and other
sources. The FNL reanalysis data are made with the same model as NCEP uses in the Global
Forecast System (GFS), but the FNL reanalysis data are prepared about an hour or so after the



GFS is initialized. The FNL reanalysis data parameters include surface pressure, sea level
pressure, geopotential height, temperature, sea surface temperature, soil values, ice cover,
relative humidity, winds, vorticity etc. The data temporal range for 1-degree is from July 30,
1999 to the present, while the time range for the 0.25-degree is from July 8, 2015 to the present
(https://rda.ucar.edu/datasets/ds083.2/, https://rda.ucar.edu/datasets/ds083.3/).
***The fifth generation ECMWF reanalysis (ERA5) data.*** The ERA5 is the fifth generation
ECMWF reanalysis of the global climate. Reanalysis combines model data with observations
worldwide to form a globally complete and consistent dataset. ERA5 replaces its predecessor,
the ERA-Interim reanalysis. ERA5 data is available from 1959 to present with a resolution of
0.25 º × 0.25 º (atmosphere) and 0.5 º × 0.5 º (ocean waves). The model requires 3D data and
2D DATA, and the variables of 3D data are temperature, U and V components of wind,
geopotential height, relative humidity
(https://cds.climate.copernicus.eu/cdsapp#!/dataset/reanalysis-era5-pressure-
levels?tab=overview). The 2D data mainly includes the parameters surface pressure, mean sea
level pressure, skin temperature, 2-m temperature, 2-m relative humidity, 10-m U and V
components of wind, soil data and soil height
(https://cds.climate.copernicus.eu/cdsapp#!/dataset/reanalysis-era5-single-
levels?tab=overview).
**2.1.2 Underlying surface data**
The default underlying surface data in WRF are USGS and MODIS data, where USGS has 24
classifications and MODIS has 20 classifications. In this study, MODIS data is selected. The
basic land cover is a modified IGBP (International Geosphere Biosphere Programmer), which
is calculated by supervised classification using MODIS Terra and Aqua reflectance data, with
a resolution of 500 m
(https://www2.mmm.ucar.edu/wrf/users/download/get_sources_wps_geog.html). The dataset
that comes with WRF is based on the year 2001(Bhati & Mohan, 2016). The 20 types are
evergreen needleleaf, evergreen broadleaf, deciduous needleleaf, deciduous broadleaf, mixed
forest, closed shrublands, open shrublands, woody savannas, savannas, grasslands, permanent
wetlands, croplands, urban and built-up, cropland mosaics, snow and ice, bare soil and rocks,
water bodies, wooded tundra, mixed tundra and barren tundra.





To consider the influence of the underlying surface data on the model results, we further select
the same underlying surface data as the simulation period (i.e., January 2016)
(https://e4ftl01.cr.usgs.gov/MOTA/MCD12Q1.006/). This data is MCD12Q1 version 6 data
product(Friedl et al., 2002), including 17 land types that cover the IGBP land cover
classification.
**2.2 Description of the modelling experiments**
To evaluate the effect of these uncertainties on the simulation results of the meteorological fields, a
total of 12 experiments are conducted, and the detailed configuration of the experiments is shown
in Table 1. The effect of horizontal resolution is presented by three experimental comparisons in
Exp1, Exp2 and Exp3, and the effect of vertical resolution by Exp3 and Exp4. The implications of
the surface layer schemes are analyzed by comparing three experiments in Exp5, Exp6 and Exp7.
The impact of the initial field and boundary conditions are compared by three experiments, i.e.,
Exp3, Exp8 and Exp9. The influences of the underlying surface are displayed by two Exp3 and
Exp10. The update of the model version is compared by Exp11 and Exp12.
Table 1 detail parameters setting of the experiments

| Experiments | Horizontal resolution | Vertical resolution | PBL schemes | Surface layer schemes | Initial field and boundary condition | Underlying surface | Version of Model |
|---|---|---|---|---|---|---|---|
| Exp1 | **75 km** | 48 levels | YSU | MM5 | FNL-1 ° | Modis-15s | WRF v3.9.1 |
| Exp2 | **15 km** | 48 levels | YSU | MM5 | FNL-1 ° | Modis-15s | WRF v3.9.1 |
| Exp3 | **3 km** | 48 levels | YSU | MM5 | FNL-1 ° | Modis-15s | WRF v3.9.1 |
| Exp4 | 3 km | **62 levels** | YSU | MM5 | FNL-1 ° | Modis-15s | WRF v3.9.1 |



| Exp5 | 3 km | 48 levels | **BL** | **MM5** | FNL-1 º | Modis-15s | WRF v3.9.1 |
|---|---|---|---|---|---|---|---|
| Exp6 | 3 km | 48 levels | **MYJ** | **Eta** | FNL-1 º | Modis-15s | WRF v3.9.1 |
| Exp7 | 3 km | 48 levels | **BL** | **Eta** | FNL-1 º | Modis-15s | WRF v3.9.1 |
| Exp8 | 3 km | 48 levels | YSU | MM5 | **FNL-0.25 º** | Modis-15s | WRF v3.9.1 |
| Exp9 | 3 km | 48 levels | YSU | MM5 | **EC-0.25 º** | Modis-15s | WRF v3.9.1 |
| Exp10 | 3 km | 48 levels | YSU | MM5 | FNL-1 º | **Modis-15s (2017)** | WRF v3.9.1 |
| Exp11 | 3 km | 48 levels | ACM2 | MM5 | FNL-1 º | Modis-15s | **WRF v3.9.1** |
| Exp12 | 3 km | 48 levels | ACM2 | MM5 | FNL-1 º | Modis-15s | **WRF v3.6.1[+*]** |

*WRF3.6.1[+] refers to the migration of the ACM2 scheme from WRFv3.6.1 to WRFv3.9.1, ensuring
that no changes in other parameterization schemes.
**3 Results and discussion**
**3.1 horizontal resolution impact on 2-m temperature and 10-m wind speed**
The underlying surface information is crucial to the simulation of near-surface meteorological
factors. From the distribution of the underlying surface, the three different resolutions of the model
can basically capture the general information of the underlying surface (Fig. 1). The resolution of
75 km is relatively coarse, so many fine features are ignored and represented uniformly by a large
grid (Fig. 1a). The resolution of 15 km is very significantly different compared to 75 km (Fig. 1b),
and many fine characteristics (e.g., lakes, cities, etc.) are represented, very close to the features of 3
km.



| | | | | | |
|---|---|---|---|---|---|
| 1 | Evergreen Needleleaf | 6 | Closed Shrublands | 11 | Permanent Wetlands | 16 | Bare Soil and Rocks |
| 2 | Evergreen Broadleaf | 7 | Open Shrublands | 12 | Croplands | 17 | Water Bodies |
| 3 | Deciduous Needleleaf | 8 | Woody Savannas | 13 | Urban and Built-up | 18 | Wooded Tundra |
| 4 | Deciduous Broadleaf | 9 | Savannas | 14 | Cropland Mosaics | 19 | Mixed Tundra |
| 5 | Mixed Forest | 10 | Grasslands | 15 | Snow and Ice | 20 | Barren Tundra |

**Land Use**

**177**

**178**      **Figure 1. (a-g) Map of land use type in the seven nested model domains.**





Further comparative analysis of temperature and wind speed in five regions at these three resolutions
have been performed. In terms of regional distribution, all three experiments can simulate high and
low value areas of 2-m temperature, but there are differences in the degree of overestimation and
underestimation (Fig. 2). In the NCP region, the three experiments underestimate the temperature
over a similar range of regions, especially in the northwest (Fig. 2a1-c1). Experiment 1 differs more
sharply from the other two experiments in areas with more marked underlying surface variability
such as in the complex mountainous areas in the northwest, the underestimation of Exp1 is more
significant, and at the sea-land interface, the overestimation of Exp1 is more pronounced (Fig. 2a1),
because the grid resolution is too low. The number of stations overestimated by the three
experiments is 96, 128, 172, and the relative bias are 0.38%, 0.19%, 0.18%. Although the number
of stations overestimated by Exp1 is small, there are more extreme values, so the deviation is larger.
Correspondingly, the higher degree of underestimation (-0.89%) in Exp1 derives from more
minimal values and stations (N=397) as well. For the YRD region, obviously, Fig. 2a2-c2 note that
the RB of the stations vary greatly with different horizontal resolutions, especially for the
northeastern coastal of YRD region (i.e., northeast Jiangsu province) from overestimation (Fig. 2a2)
to underestimation (Fig. 2c2), and the degree of underestimation gradually decreases in the southeast
of YRD (i.e., Zhejiang and Fujian provinces). In the SB region, it is clear that Exp1 underestimates
the 2-m temperature more significantly (RB=-1.11%, N=245), with fewer stations in the Fig. 2a3,
followed by Exp2 (RB=-1.03%, N=208), and to a lesser extent by Exp3 (RB=-0.69%, N=152). The
PRD region behaves differently from other regions, with the simulation results of Exp1 showing an
underestimation (RB=-0.11%), while Exp2 (RB=0.13%) and Exp3 (RB=0.35%) an overestimation
(Fig. 2a4-c4). The variation of underlying surface between grids in the PRD region is more complex
in comparison with other regions (Fig. 1). This does not indicate that the simulation results are better
when the grid horizontal resolution is lower, because the scheme itself still has errors in the
simulation. It only reveals that the simulation results of Exp1 perform better statistically in the
current model configuration for this region. The number of stations in Exp1 in the NS region is
much less than the other two experiments, which means that the relative bias of Exp1 is more than
±3% and the deviation is greater, for the area along the Qilian mountains (Fig. 2a5-c5) in particular.
The results of wind speed are different from those of temperature, and the difference between the
three experiments is not as obvious as that of temperature (Fig. 3). The three experiments
overestimate the wind speed to varying degrees, however, more stations underestimate wind speed



in the Exp1, especially in the NCP ($N_{Exp1}$=34, $N_{Exp2}$=21, $N_{Exp3}$=19) and SB region ($N_{Exp1}$=29,
$N_{Exp2}$=18, $N_{Exp3}$=7) (Fig. 3a3). As the grid resolution is too coarse in the Exp1, the wind speed is
underestimated at some stations due to the complex terrain in the NCP and SB regions (Fig. 3a1,
a3).

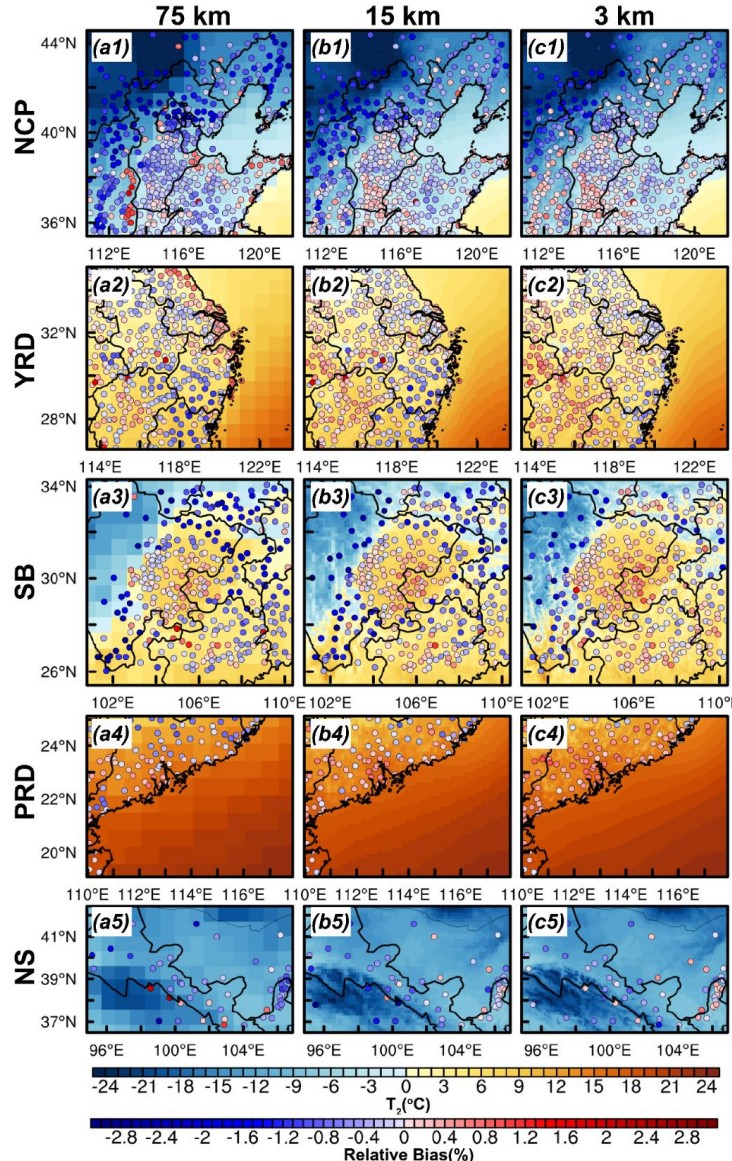


**Figure 2. Regional distribution of 2-m temperature simulated by the (a) domain 1 (75 km), (b)**
**domain 2 (15 km) and (c) domain 3 (3 km) for five regions in January, and distribution of relative**
**bias between simulations and observations is denoted by scatters.**





It can also be seen from Figure 4 that the three experiments have a large difference in temperature
simulation, and the underestimate in Exp1 is more significant (Fig. 4a1-a5). However, in the PRD
region, although the average value of the mean bias is closer to 0 on account of the offsetting positive
and negative deviations. For the distribution range of the mean bias, it has been found that the
distribution of the Exp3 is closer to 0 (Fig. 4a4). In terms of the cumulative probability distribution,
the simulations differ for different temperature segments in the NCP, SB, and NS regions. For the
NCP region, the temperature below 270 K is better simulated in Exp3, the temperature threshold in
the SB region is about 280 K, and the threshold is about 265 K in the NS region (Fig. 4b1, b3, b5).
In the YRD region, the simulations of all three experiments are almost the same for any segmented
temperature (Fig. 4b2). In addition, the PRD region is special, with temperature below about 285 K,
and the Exp2 simulates better (Fig. 4b4). It is worth noting that, regardless of the region, one thing
in common is that the temperature of the three experiments simulations gets closer and closer as the
temperature increases. While the difference of wind speed between the three experiments is not
obvious (Fig. 4c1-c5). The average value of the mean bias in Exp1 is closer to 0, mainly attributable
to that there are more stations with negative mean bias to offset. Wind speed and temperature behave
differently in regard of cumulative probability distributions, with increasing differences in simulated
wind speeds for the three experiments as wind speed increases (Fig. 4d1-d5). The wind speed
simulated in Exp1 is low, leading to a better performance in Exp1 for small wind speed (Fig. 4d1-
d5).



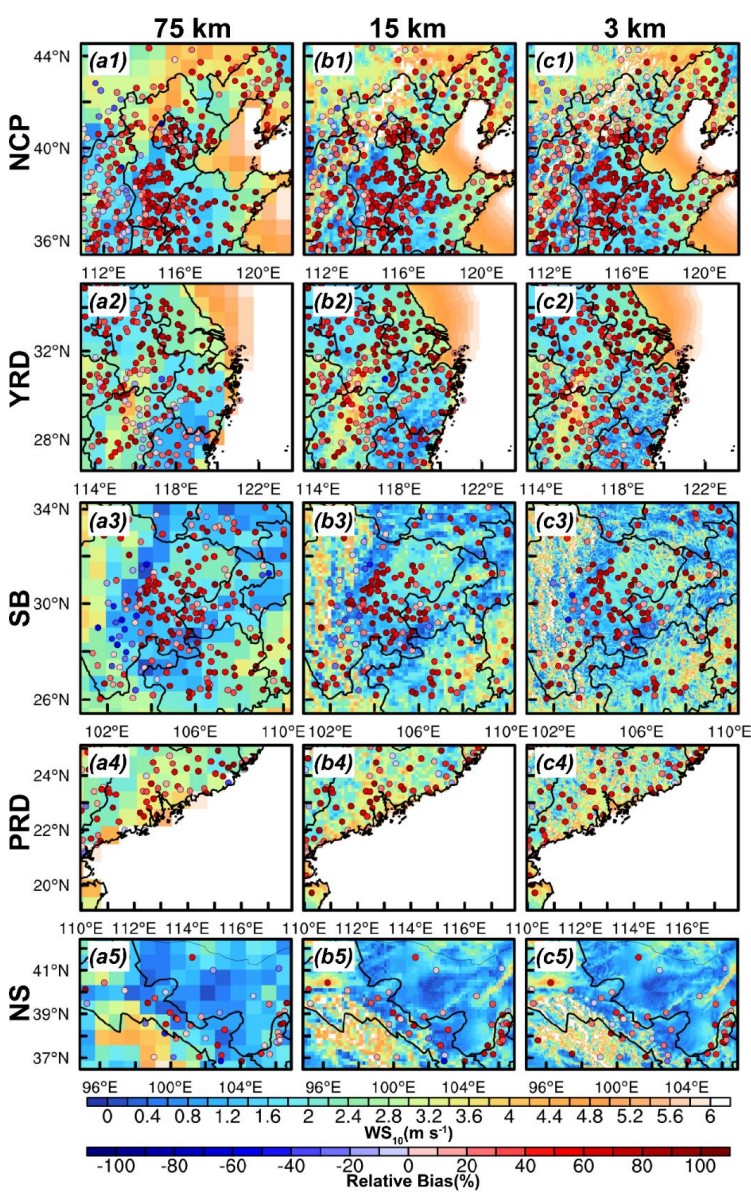


Figure 3. Similar as Figure 2, but for 10-m wind speed.

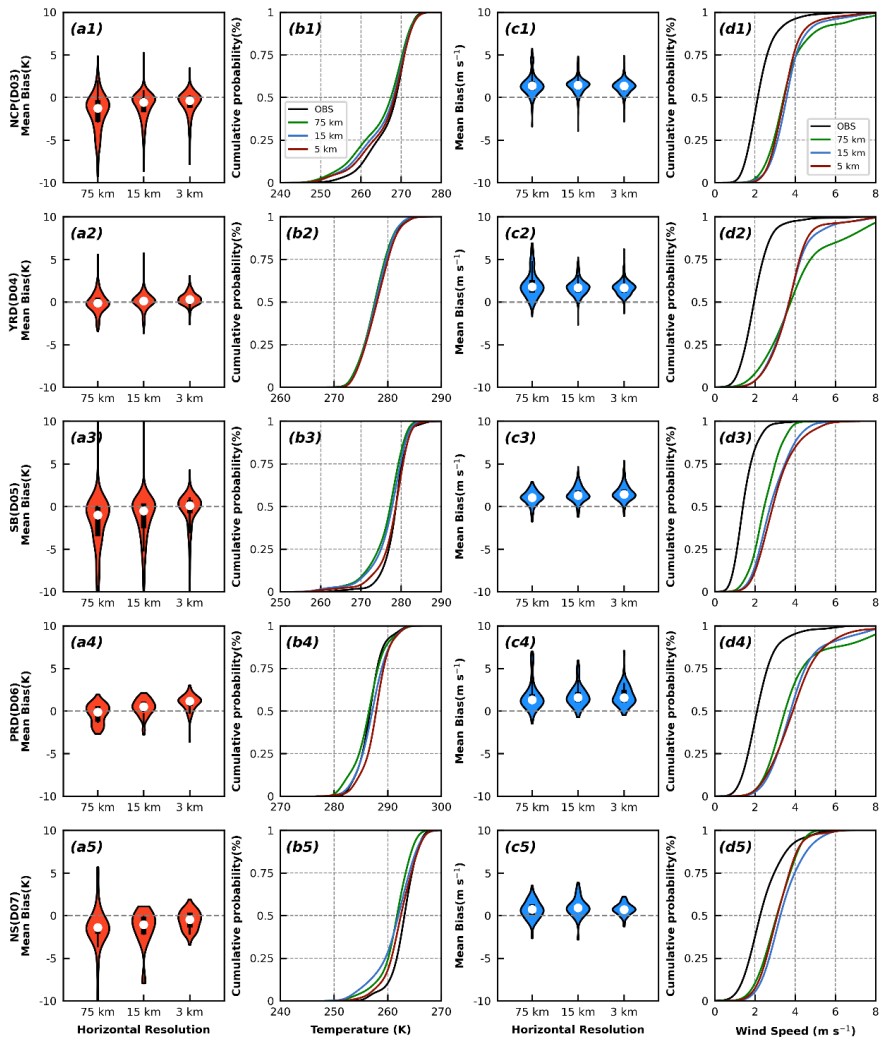

**239**

**240** Figure 4. Violin-plots of mean bias of observed and simulated (a1-a5) 2-m temperature and (c1-c5)

**241** 10-m wind speed at different horizontal resolution (i.e., 75 km, 15 km, 3 km), cumulative

**242** probability of observed and simulated (b1-b5) 2-m temperature and (d1-d5) 10-m wind speed at

**243** different horizontal resolution (i.e., 75 km, 15 km, 3 km) for five regions.

**244** **3.2 vertical resolution impact on PBL structures**

**245** Based on Exp3, the vertical resolution has been further encrypted from 21 to 35 levels below 2 km,

**246** i.e., the total number of vertical levels is increased from 48 to 62 levels (Fig. 5). The temperature

**247** and wind fields of the two experiments (Exp3 and Exp4) simulations are compared for the four



**248** sounding stations selected for each region in Part I. As can be seen from Figure 6, the re-encryption

**249** of the vertical resolution has no effect on the simulation of the temperature, regardless of the region.

**250** The simulation results of the two experiments almost overlap in the vertical direction, implying that

**251** the vertical structure of 48 levels is sufficient. On the contrary, the encryption of vertical resolution

**252** affects the simulation results of wind speed to a certain extent, but the effect is marginal, especially

**253** for high altitude regions like SB and NS (Fig. 7). For the YRD and PRD regions, the wind speed

**254** simulated in Exp4 is less than that of Exp3 below 1000 m, with a difference of less than 1 m s⁻¹.

**255** However, the encryption of the vertical resolution causes an increase in memory, which would add

**256** about 5 GB of memory for a region of 1-day results, and the 150 GB for a month. Therefore, the

**257** improvement in wind speed in some areas, due to the increase in vertical resolution, is not worth

**258** the cost of increased memory, as the improvement is simply too insignificant.

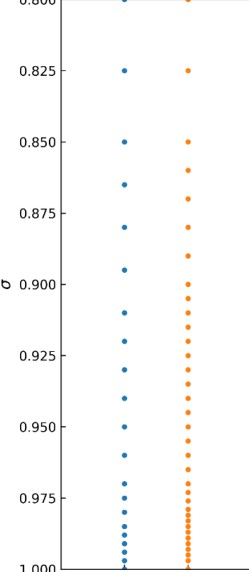

**259**
**260** **Figure 5. Vertical levels distribution for the two experiments of σ below 2 km in the model.**

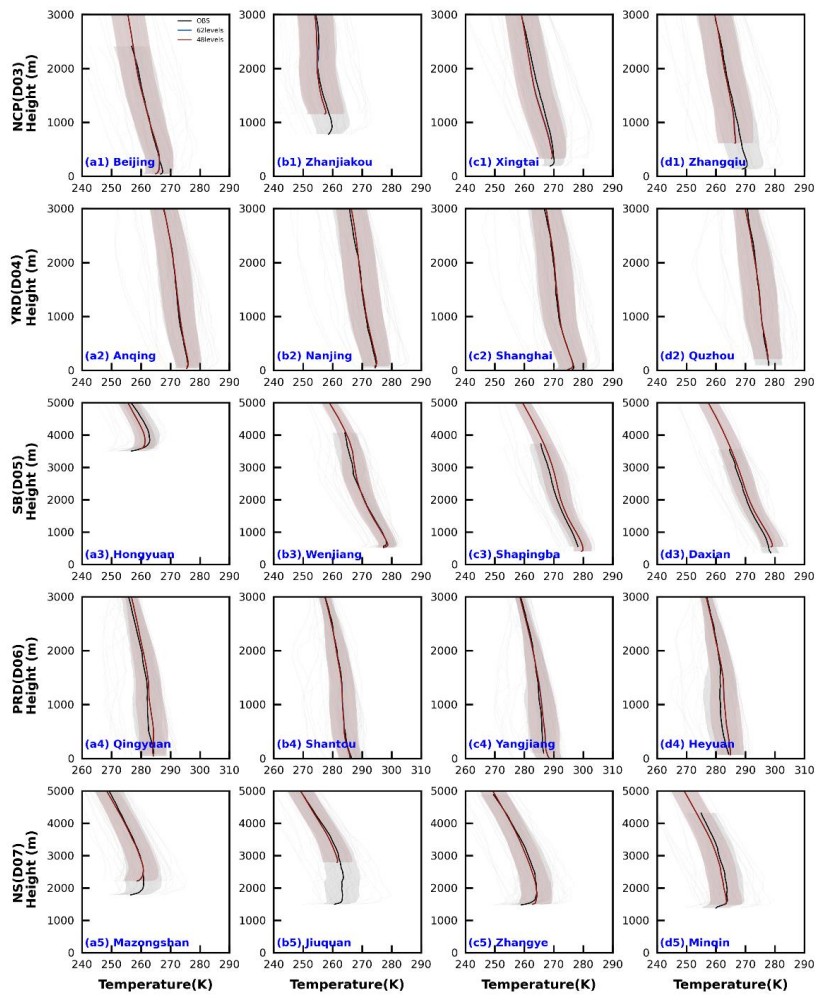

**Figure 6. Average vertical profiles of observed and simulated temperature at 08:00 and 20:00 BJT at four sounding stations for each region in January. The unobtrusive gray lines indicate the simulated lines for all time periods, and the lines with shading indicate the average values and shaded areas show the uncertainty range (the mean ±1 standard deviation).**



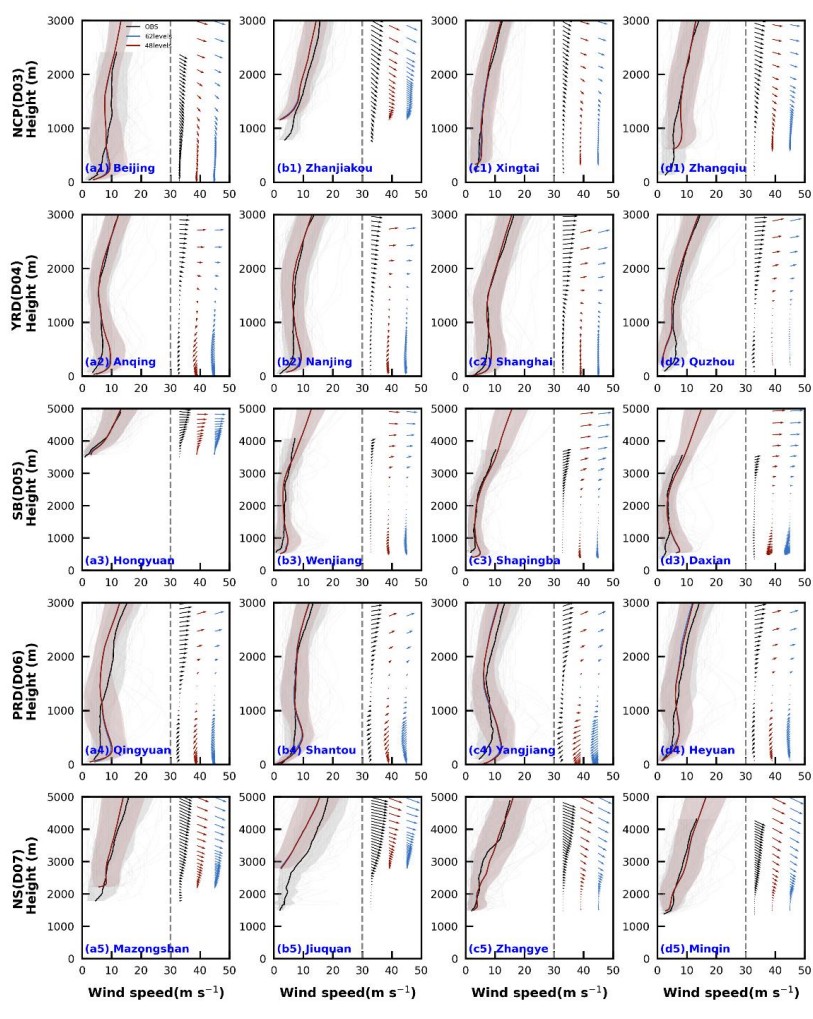

**266**

**267**   **Figure 7. Similar as Figure 6, but for 10-m wind speed and direction.**

**268**   It is not necessary to set the vertical resolution much finer compared to the horizontal resolution,

**269**   and in this experiment, 48 levels are fully sufficient to reproduce the vertical structure of the PBL.

**270**   **3.3 near-surface (N-S) scheme impact on PBL structures**

**271**   For the impact of the N-S scheme, this section focuses on the changes in the N-S meteorological

**272**   parameters.

**273**   The N-S and PBL schemes are fixed pairings, and three experiments (i.e., Exp5, Exp6 and Exp7)

**274**   are done by this study to distinguish the extent to which the N-S and PBL schemes affect the N-S

**275**   meteorological parameters (2-m temperature, 2-m relative humidity, 10-m wind speed and direction).



In terms of daily variation, the variation of temperature in the five regions is consistent, with similar
simulated results in Exp5 (BL+MM5) and Exp7 (BL+Eta), and two experiments have notable
differences from Exp6 (MYJ+Eta) (Fig. 8a1-a5). However, the relative humidity and temperature
are different, and the results of Exp5 and Exp7 are not close to each other (Fig. 8b1-b5). From the
results of wind speed, it is similar to the results of temperature, and the results of Exp5 and Exp7
are much closer, as is the wind direction (Fig. 8c1-d5). Furthermore, the three schemes are made
differential to quantify the impact of the PBL scheme and N-S scheme. Exp6-Exp7 note the impact
of the PBL scheme, and Exp5-Exp7 illustrate the effect of the N-S scheme.

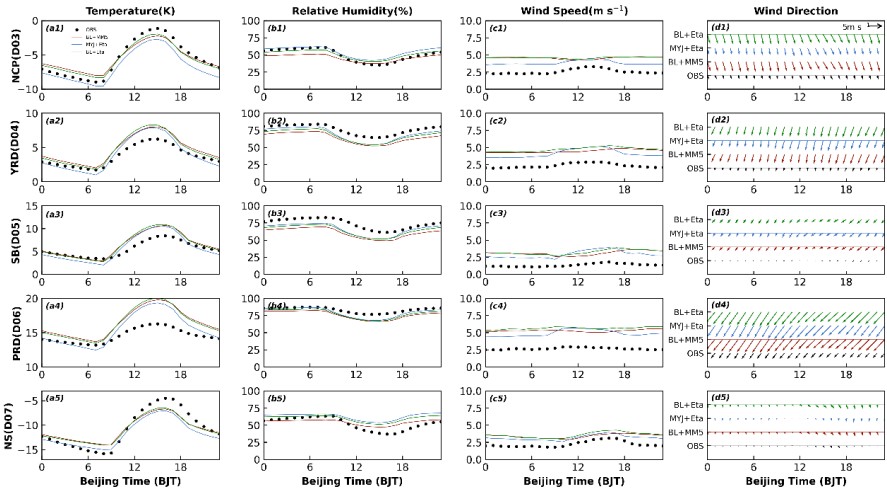


**Figure 8. Time series of diurnal variation of (a1-a5) 2-m temperature, (b1-b5) 2-m relative**
**humidity, (c1-c5) 10-m wind speed and (d1-d5) 10-m wind direction for five regions in January.**
As can be seen from Fig. 9a1-a5, the influence of the PBL scheme is greater compared to the N-S
scheme in five regions. The difference in temperature simulated by different PBL schemes is about
1 K, while the difference for N-S schemes is just less than 0.5 K. In Figure 9b1-b5, as in Figure 8,
the results for relative humidity differ from those for temperature. The PBL scheme does not affect
the relative humidity to the same extent as the N-S scheme, and it is also less than the N-S scheme.
Particularly in the NCP, SB, and NS regions, the impact of the PBL scheme is much smaller than
that of the N-S scheme (Fig. 9b1, b3, b5). Regardless of the PBL scheme and N-S scheme, the effect
is greater at night than during the day. The findings for wind speed and temperature are more similar,
with the PBL scheme having a remarkably greater impact than the N-S scheme (Fig. 9c1-c5). Except
for the daytime in both YRD and PRD regions, the N-S scheme has a slightly greater effect on wind



**297** speed than the PBL scheme (Fig. 9c2, c4). The wind direction is divided into a total of eight

**298** directions (N, N-E, E, S-E, S, S-W, W, N-W), and the influence of the PBL scheme is larger as to

**299** the percentage frequency of each direction (Fig. 9 d1-d5).

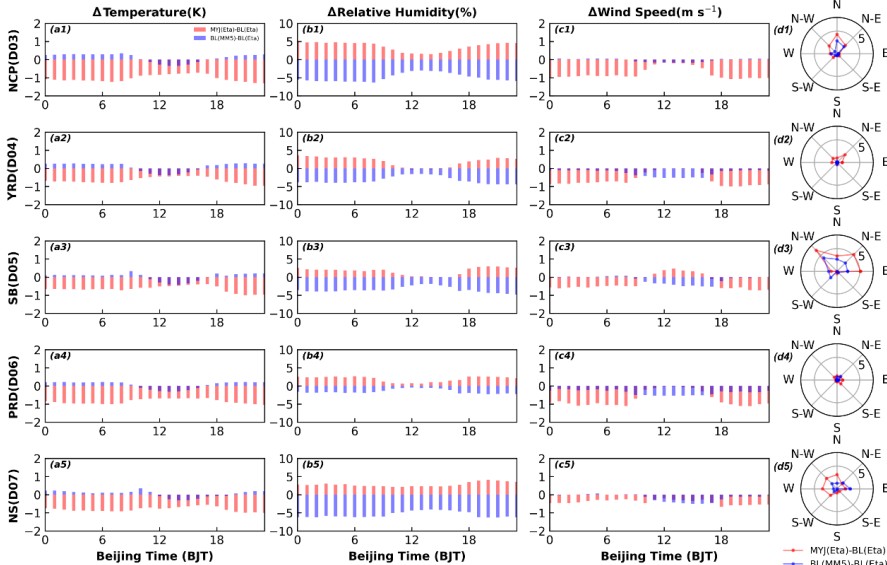

**300**

**301** **Figure 9. Time series of diurnal variation of the effects of PBL scheme and N-S scheme on (a1-a5)**

**302** **2-m temperature, (b1-b5) 2-m relative humidity, (c1-c5) 10-m wind speed and (d1-d5) 10-m wind**

**303** **direction for five regions in January.**

**304** As for the regional distribution of temperatures, the distribution of Exp5 and Exp7 is more similar,

**305** without regard to the region, and it differs considerably from that of Exp6 (Fig. 10). Therefore, for

**306** temperature, the effect of the PBL scheme is more important. For wind speed, Exp7 simulates the

**307** largest wind speed, followed by Exp5, and Exp6 has the smallest wind speed, noting that the PBL

**308** scheme has a larger degree of influence than the N-S scheme (Fig. 11).



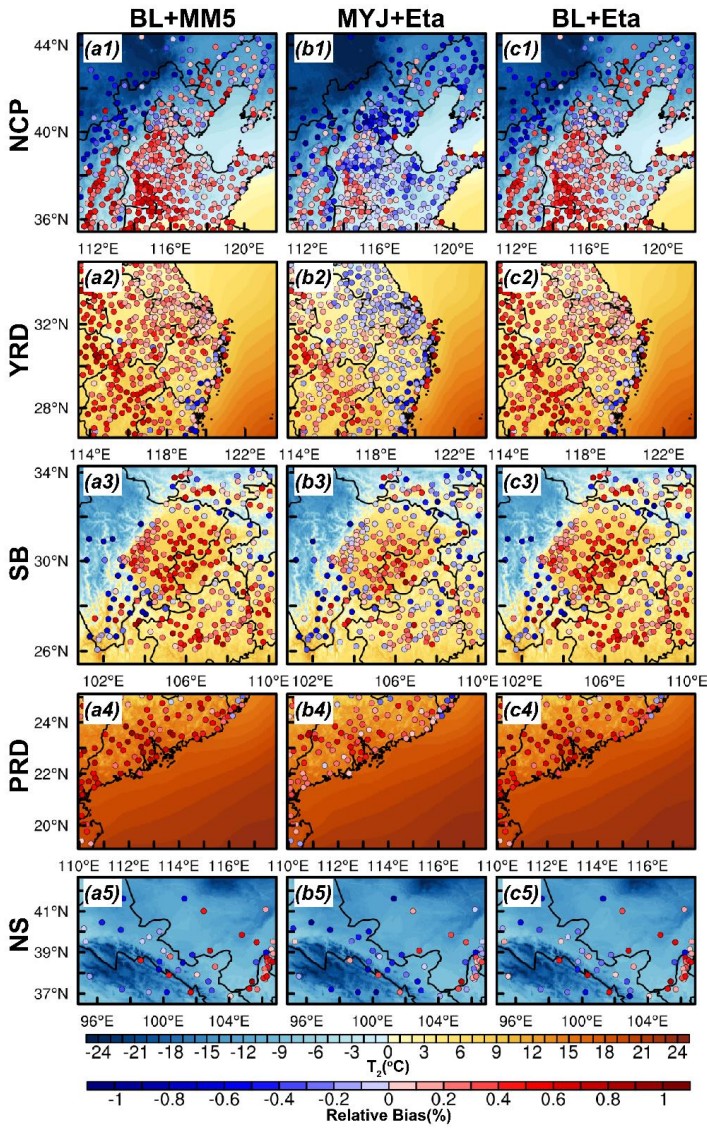

309

**Figure 10. Regional distribution of 2-m temperature simulated by the (a) BL+MM5, (b) MYJ+Eta**

**and (c) BL+Eta for five regions in January, and distribution of relative bias between simulations**

**and observations is denoted by scatters.**

In general, for temperature, the choice of PBL scheme is of much more importance. For relative

humidity, the PBL and N-S schemes are equally important, except for the NCP, SB and NS regions,

where the choice of the N-S scheme is more principal. For wind speed and direction, the choice of

PBL scheme is more critical, and the simulation of different PBL schemes leads to more differences

in the results.

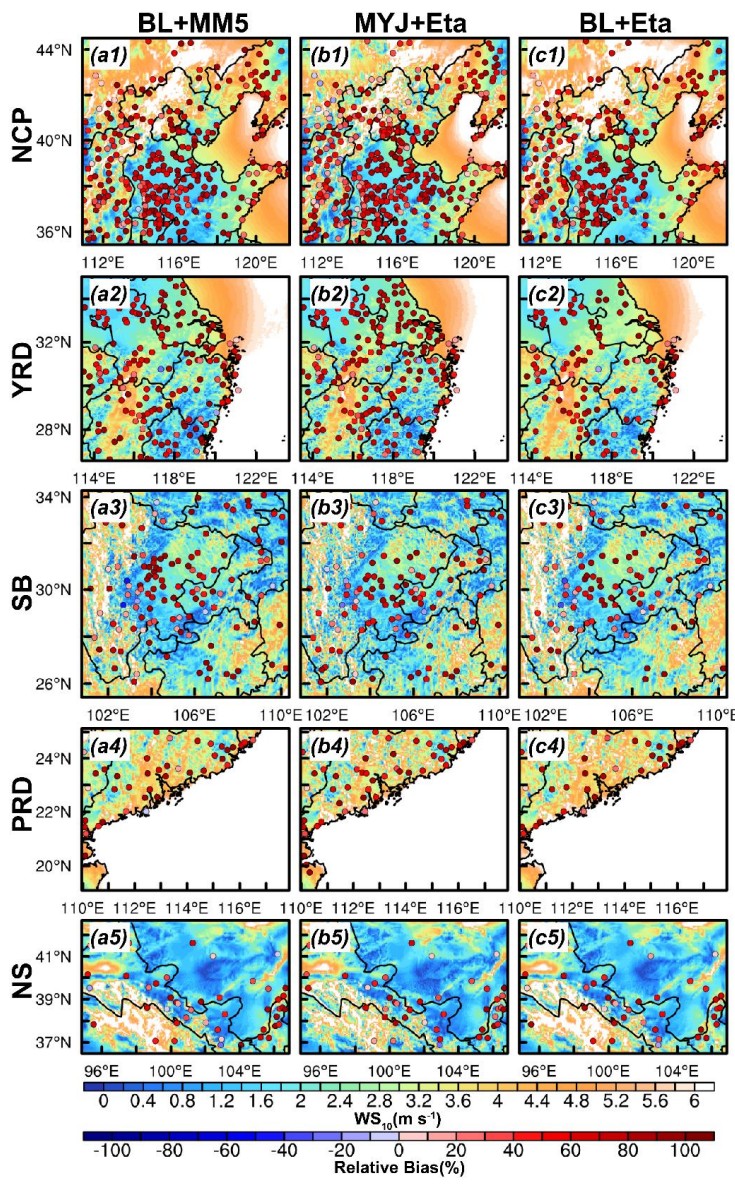


**Figure 11. Similar as Figure 10, but for 10-m wind speed.**
**3.4 effect of initial and boundary conditions on meteorological parameters**
In this subsection, the same initial field and boundary conditions at different resolutions (i.e., FNL–
1° and FNL–0.25°) and different initial field and boundary conditions at the same resolution (i.e.,



FNL–0.25° and EC–0.25°) are chosen to explore the effects of the initial field and boundary
conditions on the meteorological field simulation. Figure 12 shows the daily variation series of 2-m
temperature, 2-m relative humidity and 10-m wind speed and direction. Also, Figure 12 notes that
for temperature and relative humidity, the effect of data with different resolutions of the same initial
field on the results is small, but the effect of data with different initial fields of the same resolution
is profound. For the five regions, the EC data better simulate the temperature than the FNL data
during the day, while at night, the difference between the two types of data simulating the
temperature becomes less than during the day, except for the NCP and NS regions (where the
temperature difference is larger for both day and night) (Fig. 12 a1-a5). For relative humidity, the
EC data are simulated better than the FNL data regardless of the region, playing a key role in
improving the relative humidity results of the model (Fig. 12 b1-b5). Overall, the increase in
resolution of the initial field data from 1° to 0.25° has less effect on the simulation of temperature
and relative humidity, while there is a striking difference between the different initial field data.

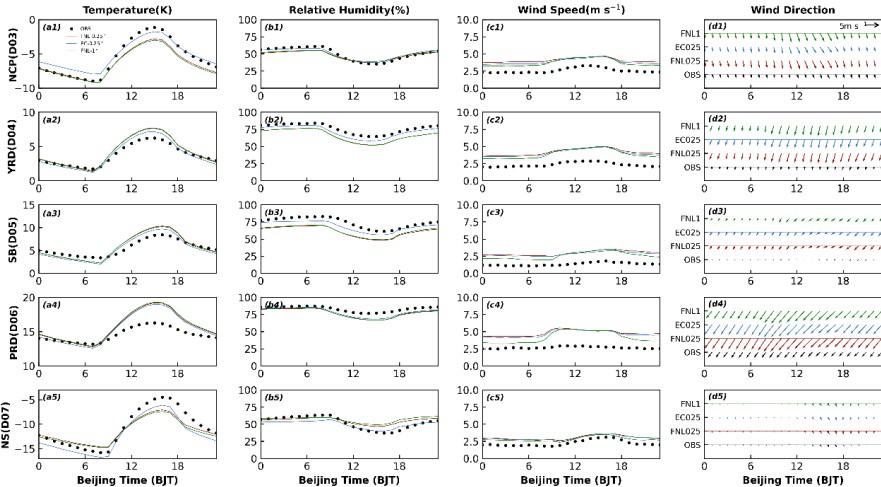


**Figure 12. Similar as Figure 8, but for different initial and boundary conditions.**
The results for wind speed differ from the first two parameters in that there is almost no difference
between the three experiments for wind speed simulations during the day (Fig. 12 c1-c5). However,
different initial field data at the same resolution have very little effect on the wind speed, but the
same initial field data at different resolutions have a significant effect on the wind speed, especially
at night (Fig. 12c1-c5). All data have a negligible effect on the wind direction (Fig. 12 d1-d5).
The EC data have improved the results of relative humidity for all regions as mentioned earlier (Fig.



12 b1-b5). In terms of regional distribution, the regional distribution of FNL data is similar in shape
for different resolutions (Fig. 13 a, c). However, the relative humidity distribution simulated by EC
data and FNL data is drastically different (Fig. 13). It is worth noting that the relative humidity of
the EC data is the highest in the four regions except for the NS region in which the relative humidity
is the lowest (Fig. 13 b1-b5).

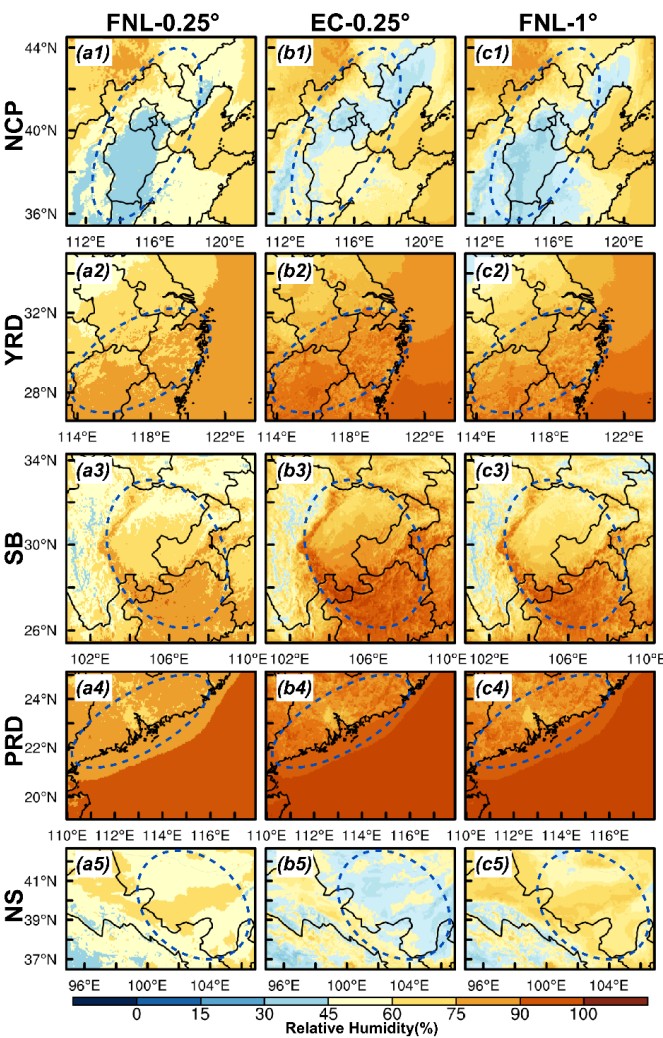


**350 Figure 13. Regional distribution of 2-m relative humidity simulated by the (a) FNL-0.25°, (b) EC-**

**351 0.25° and (c) FNL-1° for five regions in January, and the blue dashed circles indicate the regions**

**352 where the results of the three experimental simulations differ significantly.**

In the vertical direction, the simulated results of the three experiments for temperature and wind





speed do not differ much, unlike the near-surface meteorological parameters (i.e., T2, RH2, WS10
and WD10) that show such obvious differences (Fig. S1, S2). Nevertheless, for the relative humidity,
the variation in vertical direction at different heights is more consistent with the near-surface layer,
where the relative humidity of EC data is high in the whole layer (Fig. S3). Except for a few highland
stations outside the basin in the SB region, the relative humidity of EC data is low at higher altitudes
(Fig. S3 a3, b3, d3).
**3.5 effect of underlying surface on meteorological parameters**
To further explore the impact of underlying surface changes on the simulation results of
meteorological fields, we use the underlying surface data in January 2016 that is closer to the
simulation time, in addition to the default underlying surface data that comes with the model, for
comparative analysis of the simulation. Comparing Figure 1 and Figure 14, it can be concluded that
the most substantial change in the Domain 1 area is in the croplands type (i.e., code 12), especially
for the area south of latitude 30 °N. Many types with an underlying surface of 12 have become 14
or 8, 9 etc. Although both 12 and 14 here can represent cropland, there are some differences in the
specific descriptions. Code 12 mainly indicates that at least 60% of area is cultivated cropland, while
code 14 mainly refers to the mosaics of small-scale cultivation 40–60% with natural tree, shrub, or
herbaceous vegetation. In addition to croplands, the two types of urban and water bodies are more
variable as well. Therefore, this subsection focuses on the effects of urban and water body changes
on surface meteorological fields.



**373**

**374** **Figure 14. Similar as Figure 1, but for the land use type for January 2016.**






In terms of the overall regional distribution, the new underlying surface did not affect the areas of
high and low values of temperature (Fig. 15 a-b) to an important degree. However, the difference
between the simulation results of two different underlying surface shows that the change of the
underlying surface has an effect on the temperature by about ±1 ℃, especially for the grids with
more obvious changes in water bodies and urban areas (Fig. 15 c). In the NCP region, an increase
in the area of water bodies in the coastal areas of Tianjin, Shandong Peninsula, and Jiangsu Province
leads to a distinct increase in temperature (i.e., indicated by red boxes), while a decrease in the area
of inland water in the northern region of Shanxi Province causes a decrease in temperature (i.e.,
denoted by blue boxes) (Fig. 15 c1). The decrease in the area of water bodies in the Yangtze River
in the YRD region has caused a decrease in temperature, while urbanization has contributed to an
increase in temperature in several regions (Fig. 15 c2). The underlying surface changes in the SB
region are mainly in the form of forest and savannas changes, as well as the more rapid urbanization
of the provincial capital city of Chengdu (Fig. 14 e). The development of this city has a positive
feedback effect on the temperature of the region (Fig. 15 c3). The underlying surface change in the
YRD region is from croplands to savannas, with a rapid greening rate, and its excessive greening
may make the green coverage of some cities too high, leading some grids to identify the cities as
savannas. In the NS region, the area of croplands and cities along the Qilian Mountains increases
and the area of some inland lakes decreases, in turn leaving some influence on the results of the
temperature.
The wind field does not vary as regularly as the temperature filed. Except for the variation of water
body area which has a more consistent pattern on the wind field, all other types of underlying surface
variation have a haphazard effect on the wind field (Fig. S4).



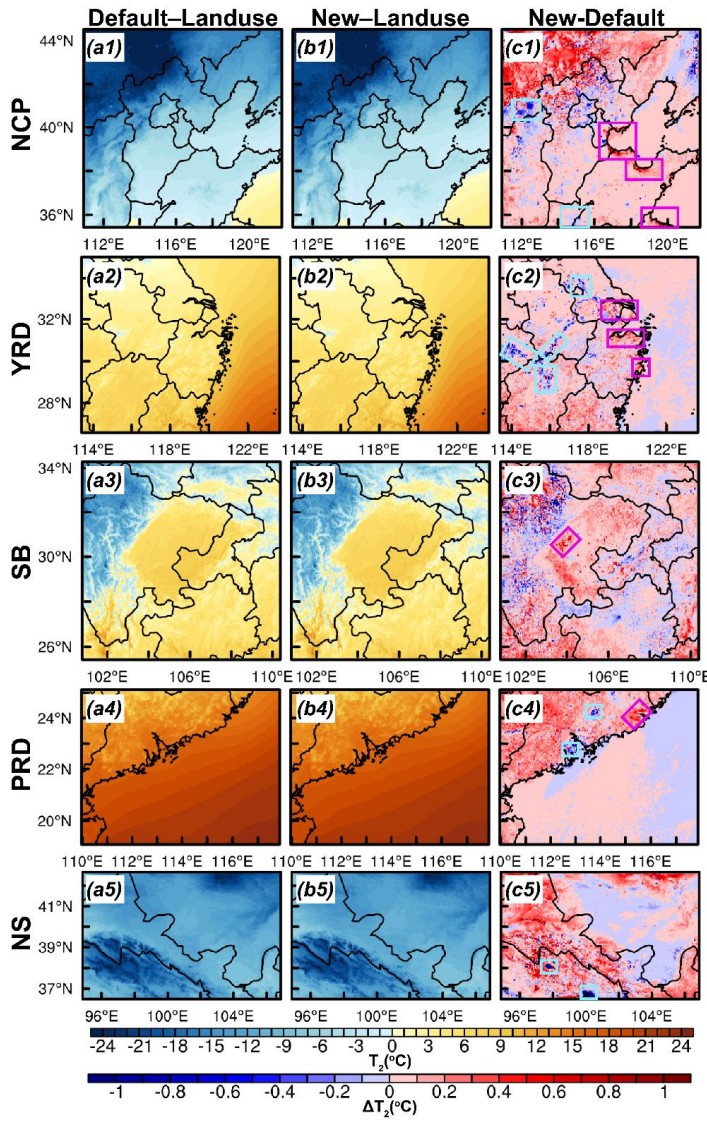

**Figure 15. Regional distribution of 2-m temperature simulated by the (a) default land use, (b) new land use and (c) the difference between the two land use types for five regions in January. The blue (red) box indicates the region where the wind speed decreases (increases) due to changes in the water bodies and urban.**

### 3.6 impact of the model version update

As computer technology continues to evolve, the parameterizations in the model are being upgraded





and improved, but it is worthwhile further exploring how much the parameterizations and versions
affect the simulation results of the model. For the PBL parameterization scheme, turbulent diffusion
is crucial for the vertical mixing of momentum, heat, water vapor and pollutants within the PBL.
And in December 2014, the ACM2 parameterization scheme received two major updates: (1) The
turbulent diffusion coefficients of heat are updated. The stability function of Richardson number is
modified, expecting to reduce the day and night 2-m temperature bias. (2) The bug that the minimum
value of the PBLH is lower than the height of the first level of the model under stable conditions
has been restored, and the minimum value of the PBLH is fixed to the height of the first level of the
model. We, therefore, choose the ACM2 scheme in WRFv3.6.1 as a control experiment. In the
control experiment, the ACM2 scheme in the WRFv3.9.1 version is replaced with WRFv3.6.1, and
all other schemes are kept in the WRFv3.9.1 (i.e., WRFv3.6.1+). This ensures that the difference
between the two experiments is the representative of the impact of the ACM2 scheme update.

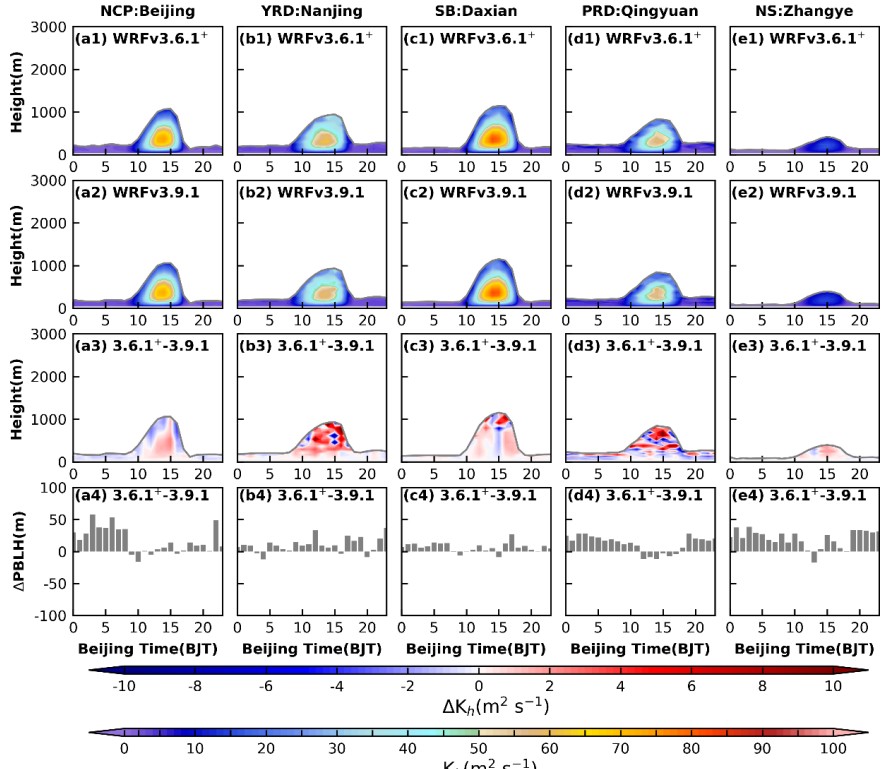


**Figure 16. Time-height cross sections of heat turbulent diffusion coefficient (TDC) simulated by**
**(a1-e1) WRFv3.6.1+, (a2-e2) WRFv3.9.1, (a3-e3) the difference between the TDC of the two versions.**



**420** (a4-e4) Time series of diurnal variation of the difference between the PBLH of the two versions.

**421** The gray line in (a1-e3) indicates the PBLH.

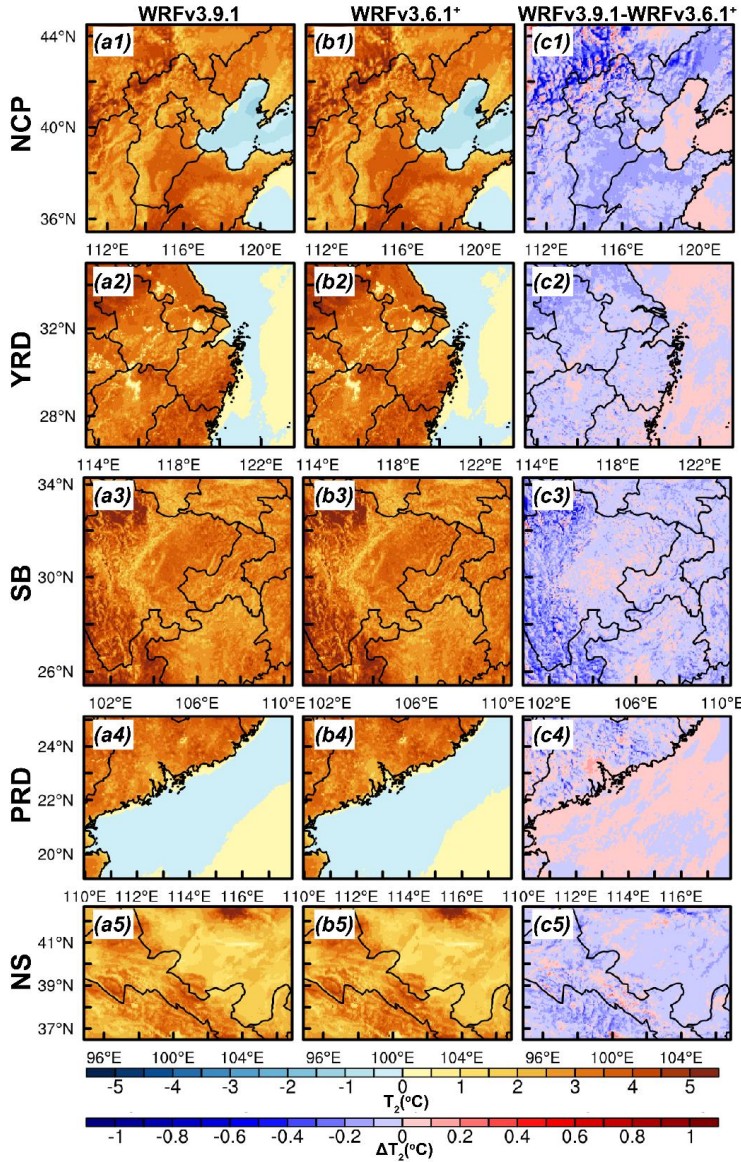

**422**

**423** Figure 17. Similar as Figure 15, but for the effects of model versions.

**424** The difference between the turbulent diffusion coefficient of heat calculated by the two versions lies

**425** in the different principles of calculation using the Richardson number (Ri) method. In the

**426** WRFv3.6.1+, not only Ri is used to judge the stability, but also z/L is used to additionally constrain





the stability and determine the empirical stability function. In contrast, only Ri is adopted to
determine the function in the WRFv3.9.1. Figure 16 shows the diurnal variation of turbulent
diffusion coefficient of heat with height, as well as the difference of PBLH. In general, the two
versions have no effect on the overall trend of TDC (Fig. 16 a1-e2). However, within the PBL, the
TDC of WRFv3.9.1 is smaller than that of WRFv3.6.1+, with the most significant difference during
the daytime. Meanwhile, in some regions at night, a TDC of WRFv3.9.1 is also greater than that of
WRFv3.6.1+ (Fig. 16 a3-e3). Besides, the differences among the five regions slightly vary. The
deviation in the NCP, SB, and NS regions is small, around 2 $m^2$ $s^{-1}$ (Fig. 16 a3, c3, e3), while the
deviation in the YRD and PRD regions is large, up to about 10 $m^2$ $s^{-1}$ (Fig. 16 b3, d3). The TDC
modification aims to reduce the temperature difference between day and night. Indeed, this
expectation is fulfilled. It can be noticed in Figure 17, the diurnal temperature difference for
WRFv3.9.1 is smaller than that of WRFv3.6.1+ in almost all regions (except for the area where the
underlying surface is water). In addition, we need to pay attention to the variation of the PBLH. As
shown in Figure 16, the difference in PBLH during daytime is smaller than at night, and the PBLH
of WRFv3.9.1 is lower than that of WRFv3.6.1+. The model WRFv3.9.1 fixes the minimum value
of the PBLH to the first level height, markedly reducing the PBLH at night. But this approach may
be too crude and parsimonious to cause problems, and should be corrected in the future.
**4 Conclusion**
The simulation results of the model within the PBL are subject to many factors, but its portrayal and
description by the PBL parameterization schemes plays a vital role in affecting the variation of the
meteorological field. The simulations of the PBL schemes on the meteorological fields has been
described in Part I. In Part II, further uncertainties affecting the results of the meteorological field
are evaluated and analyzed, and the degree of influence of different factors is compared, hoping to
provide a reference for scholars conducting research on the model.
In addition to the dominant role of the PBL scheme on the results of the meteorological field, many
elements in the model are influenced by large uncertainties. For example, what is the effect of
horizontal resolution, and how much does the result vary under different resolution conditions? Is
the continuous encryption of the vertical levels necessary for the simulation of the vertical structure
of the PBL? Which has a greater impact on the results of the meteorological field, the near-surface
(N-S) layer scheme or the PBL scheme? How much is the impact of these changes on the underlying



**457**    surface, which is constantly updated by the development of urbanization? The innovation of

**458**    computer technology has brought the opportunity to keep the model being updated. How much

**459**    effect will the updates have on different versions of the model results? The simulation of the model

**460**    depends on the initial and boundary conditions, so how much does the initial and boundary

**461**    conditions of different resolutions and products affect the model results? These uncertainties have

**462**    not been fully evaluated and analyzed yet. To resolve the confusions, this study synthesizes the

**463**    effects of the above factors on the model results.

**464**    a. ***Effect of the horizontal resolution.*** The three different resolutions have a more dramatic effect

**465**      on temperature than on wind speed. Regardless of the region, the distribution of temperature

**466**      deviations simulated at 75 km resolution is clearly different from that of 15 km and 3 km,

**467**      especially in areas with more complex topography, such as NCP, SB and NS regions. All three

**468**      resolutions overestimate the wind speed in all regions, except for the 75 km resolution, where

**469**      there is an underestimation of the wind speed at the stations around the basin in the SB region.

**470**      The difference between the resolutions decreases with increasing temperature, but becomes

**471**      more pronounced with increasing wind speed.

**472**    b. ***Effect of the vertical resolution.*** The number of vertical levels of the model is encrypted from

**473**      48 to 62 levels, with almost no effect on the vertical structure of the PBL. Meanwhile, the

**474**      increase in the number of vertical levels brings into an increase in memory of about 150 GB for

**475**      one month. Compared to the horizontal resolution, the vertical resolution does not need to be

**476**      set particularly fine, and 48 levels are perfectly sufficient to reproduce the evolution of the PBL

**477**      structure.

**478**    c. ***Influence of the N-S scheme.*** The PBL scheme makes a greater impact on the simulated results

**479**      for temperature, wind speed and direction, while for relative humidity, the N-S scheme

**480**      contributes largely, especially in the NCP, SB and NS regions. For either scheme, the effect is

**481**      much greater at night than during the daytime. In general, the choice of the PBL scheme is more

**482**      critical for temperature and wind fields. But for relative humidity, the PBL and N-S schemes

**483**      are equally important.

**484**    d. ***Impact of the initial and boundary conditions.*** The effect of data of different resolutions of the

**485**      same product on the results of temperature and relative humidity is small, but the influence of

**486**      data of different products of the same resolution is large. EC data simulates temperature better

**487**      than FNL data during the daytime, while at night, the difference between the two data is



relatively small (except for the NCP and NS regions). The EC data simulate the relative
humidity better than the FNL data regardless of the region, even in the vertical direction, which
will exposes a key way to improve the relative humidity results of the model in the future.
Nonetheless, data of the same resolution but different products exhibit no obvious effect on
wind speed, while the influence of data from the same product with different resolutions is
larger, especially at night.
*e.*   ***Effect of the underlying surface.*** In terms of regional distribution, the new underlying surface
make no significant difference with respect to the temperature. However, for the grids with
more pronounced changes in water bodies and urban, the change in underlying surface has an
approximate ±1℃ influence on temperature. An increase (decrease) in the area of water bodies
leads to an increase (decrease) in temperature, and the growth of urbanization brings about an
increase in temperature. The variation of wind field is not as regular as temperature. Except for
the changes in the area of water bodies that affect the wind filed consistently, other types of
underlying surface changes show a haphazard effect on the wind filed.
*f.*   ***Influence of the model version.*** The update of the PBL scheme reduces the day and night 2-m
temperature bias. But the simple definition method of fixing the minimum value of the PBLH
as the first level height of the model may have some defects. The change in the stability function
of the Richardson number alters the turbulent diffusion coefficient of heat, which is more
distinct in the daytime with a deviation of about 10 $m^2$ $s^{-1}$.
In summary, the horizontal resolution is more influential than the vertical resolution. The N-S
scheme has less effect than the PBL scheme on the results of temperature and wind speed. Also,
the initial and boundary conditions of different products have the most significant influence on
relative humidity. Grid changes where the underlying surface is urban and water bodies have a
more pronounced effect on the results of meteorological fields, especially for temperature. The
constant updating of the model version does not necessarily optimize the simulation results
continuously. A special advice here is that the needs of different scholars for the model vary a
lot, thus, the configuration of uncertainties requires a comprehensive consideration to obtain
the optimal results for the analysis.
**Code and data availability**
The source codes of WRF version 3.9.1 and 3.6.1 can be found on the following website:
https://www2.mmm.ucar.edu/wrf/users/download/. The original model settings file is already



included in Supplement in Part I, while the other model settings file used in Part II is named after
the file name "L62_namelist.input" and is already included in Supplement. In addition, the
observations used are also provided in Supplement in Part I. The initial field and boundary condition
data and the underlying surface data are provided in the text.

## Author contributions

Development of the ideas and concepts behind this work was performed by all the authors. Model
execution, data analysis and paper preparation were performed by WJ. XZ and HW provide
computing resources, and offer advice and feedback. YW, DW, and JZ support the data. WZ, LZ,
LG, YL, JW, YY, and YL provides suggestions. All authors contributed to the manuscript.

## Competing interests

The authors declare that they have no conflict of interest.

## Acknowledgements.

The work was carried out at the National Supercomputer Center in Tianjin, and the calculations
were performed on TianHe-1 (A).

## Financial support

This research is supported by NSFC Major Project (42090031), NSFC Project (U19A2044).

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
