# Peer review of "Comprehensive evaluation of typical planetary boundary"

_Geoscientific Model Development, 2023_

## Author Comment (AC1)

**Response to Referee #1**

**RE:** Comprehensive evaluation of typical planetary boundary layer (PBL) parameterization schemes in China. Part II: Influence of uncertainty factors

**Author(s):** Wenxing Jia et al.

By using 12 sensitive experiments, this manuscript analyzes comprehensively and quantitatively the influence of six uncertainties (including horizontal resolution, vertical resolution, initial and boundary conditions, underlying surface update, near-surface scheme, and update of the model version) on the model simulated meteorological fields. Overall, the manuscript is informative and is a good reference for model users. A few suggestions are as follows.

Thank you for your positive comments and valuable suggestions to improve the quality of our manuscript. Based on these comments and suggestions, we have made careful modifications to our pervious draft, and the detailed point-by-point responses are listed below.

**Specific comments:**

(1) Abstract: The Abstract may give more detailed information about uncertainties, such as different products of initial and boundary conditions, horizontal and vertical resolution, etc.

**Re1:** Revised as suggested. We have modified the abstract.

(2) Introduction: The logical flow of the introduction can be improved to provide a clearer view of the model uncertainties research progresses.

**Re2:** Revised as suggested. We have revised the introduction.

(3) Data and Methodology: The model settings need to be introduced in more detail. Are they the same as the settings in Part I?

**Re3:** Revised as suggested. As per your comments, we have refined this section and it is indeed the same as Part I's settings.

(4) A figure summering the influence of different factors may be added to the conclusion section, and it will allow the readers to better understand the major findings of this study.

**Re4:** Based on your suggestion, we have simply added a graph to indicate the degree of influence of different factors.

[Figure]

**Figure R1.** An overview figure of the prioritization of uncertainties, where the uncertainties are in black font and the elements focused on in that factor are in blue font.

**Technical Corrections**

1. Abstract, L41, the full name of the "EC" in "EC data" should be given.

   **Re1:** Revised as suggested.

2. L48, it may be more appropriate to change "elements" to "parameters", which are used more often.

   **Re2:** Revised as suggested.

3. L49, "structure" should be changed to "structures".

   **Re3:** Revised as suggested.

4. L50, "scheme" should be changed to "schemes".

   **Re4:** Revised as suggested.

5. L51, what are the "different parameters, regions and seasons" included, please indicate.

   **Re5:** Revised as suggested.

6. L76, "higher" should be changed to "finer".

   **Re6:** Revised as suggested.

7. L80, lack of reference support.

   **Re7:** Revised as suggested, References have been added.

8. L94-96, further clarification on other initial and boundary condition data.

   **Re8:** Revised as suggested.

9. L100-101, lack of reference support.

   **Re9:** Revised as suggested.

10. L115, "(FNL)" should be deleted.

    **Re10:** Revised as suggested.

11. L126-127, "ECMWF" should be changed to "EC".

    **Re11:** Revised as suggested,

12. L131, "DATA" should be lowercase.

    **Re12:** Revised as suggested.

13. L142, the acronym "IGBP" should be placed in parentheses.

    **Re13:** Revised as suggested.

14. L186, "and" should be changed to "but".

    **Re14:** Revised as suggested.

15. L191-192, the full name of the "N" and "RB" should be given.

**Re15:** Revised as suggested.

16. L220, "although" should be deleted.

**Re16:** Revised as suggested.

17. L221-222, this sentence needs to be reconsidered.

**Re17:** Revised as suggested.

18. L248, the four stations in each region should be represented again.

**Re18:** Revised as suggested.

19. L381, "Tianjin" should be changed to "Tianjin Shi", and three regions are best marked in the Figure 15.

**Re19:** Revised as suggested.

20. L388, "Chengdu" should be changed to "Chengdu Shi".

**Re20:** Revised as suggested.

21. L426-441, "WRFv3.6.1+" should be changed to "WRFv3.6.1+".

**Re21:** Revised as suggested.

22. L437-439, this sentence needs to be reconsidered.

**Re22:** Revised as suggested.

23. Plots in Figures 17a-b are not temperatures but temperature differences.

**Re23:** Figures 17a-b show the diurnal temperature range (DTR). Revised as suggested.

24. The quality of the images may be improved.

**Re24:** Revised as suggested, and we checked all the figures.

25. The citation format of the references needs to be corrected according to the journal's requirements.

**Re25:** Revised as suggested, and we checked all references.

---

## Author Comment (AC2)

**Response to Referee #2**

**RE:** Comprehensive evaluation of typical planetary boundary layer (PBL) parameterization schemes in China. Part II: Influence of uncertainty factors

**Author(s):** Wenxing Jia et al.

This manuscript represents an informative contribution to modelling science within the scope of the GMD, which provides a good reference for atmospheric modeling research community. the scientific quality of the manuscript is generally good with the valid scientific approach and applied methods in sensitivity modeling experiments of WRF. Some comments and suggestions are as follows:

Thank you for your positive comments and valuable suggestions to improve the quality of our manuscript. Based on these comments and suggestions, we have made careful modifications to our pervious draft, and the detailed point-by-point responses are listed below.

**Specific comments:**

(1) The most evaluations of PBL simulation are only for the near-surface meteorology, not for the entire PBL meteorology. Please clarify this limitations for this study.

**Re1:** We consider it a limitation to evaluate only the near-surface meteorological parameters and not the entire PBL. Due to the limitation of observational data and super-computing resources, many studies are conducted only for individual cases in individual regions, and the results lack generalizability. This study (i.e., Part II) and the Part I provide a more comprehensive evaluation and uncertainty analysis of the PBL schemes.

(2) The modeling validations of vertical resolution below 2000 m are compared with the meteorological sounding data. How is the vertical resolution below 2000 m for the meteorological sounding data? if the vertical resolution of sounding data is too coarse

for the both sensitivity experiments from 21 to 35 levels below 2000 m, please modify the relative conclusions on the (larger?) effect of vertical resolution.

**Re2:** For the model where the whole level is 48 levels, it is 21 levels below 2 km, while 62 levels correspond to 35 levels below 2 km. In the results of our analysis, it is found that the increase in the number of levels of the model from 48 to 62 levels does not improve the simulation significantly, while there is a smaller improvement in wind speed in individual regions. However, it is better not to need this improvement relative to the increased in memory.

(3) Please clarify the vague and misleading conclusion that the update of the model version does not necessarily optimize the model results. The updated model version can improve the meteorological simulations, and the updated near-surface scheme and PBL scheme could necessarily optimize the modeling results of .near-surface and PBL meteorology?

**Re3:** We are very sorry about the statement that there might be a problem here. We have corrected this.

(4) Please correct thoroughly the English language and usage, making the scientific results and conclusions in a clear and concise way.

**Re4:** Revised as suggested. We have reworked the entire manuscript.

---

## Author Response (AR2)

**Response to Referee #2**

**RE: Comprehensive evaluation of typical planetary boundary layer (PBL) parameterization schemes in China. Part II: Influence of uncertainty factors**

**Author(s): Wenxing Jia et al.**

**Q1:** My only one question is about the vertical resolution of the meteorological sounding data below 2000 m. The modeling validations of vertical resolution are compared with the meteorological sounding data below 2000 m. If the vertical resolution of sounding data below 2000 m is too coarse for the both sensitivity experiments from 21 to 35 levels below 2000 m, please modify the relative conclusions on the effect of vertical resolution on modeling.

**R1:** Thank you for your positive comments and valuable suggestions to improve the quality of our manuscript. It is true that the two vertical resolution settings, 21 and 35 levels, are coarse compared to the observed data, but the uncertainty analysis here focuses on how much further refinement of the vertical resolution in the model affects the uncertainty in the model results. And in the process of comparing the results of the model with the observations, the heights of the selected observations and the heights of the model layers are in one-to-one correspondence. It is certainly possible to set a finer vertical resolution in the model, which is undoubtedly a great challenge for computer resources. According to the

current conclusion, the addition of 14 levels below 2000 m does not have a great impact on the improvement of the model results, but the computational resources and data storage have increased a lot compared with the previous ones. Therefore, in the results of the current sensitivity experiments, the conclusion is applicable to the present comparison.